# High Myopia and the Complement System: Factor H in Myopic Maculopathy

**DOI:** 10.3390/jcm10122600

**Published:** 2021-06-12

**Authors:** Enrique García-Gen, Mariola Penadés, Salvador Mérida, Carmen Desco, Rafael Araujo-Miranda, Amparo Navea, F. Bosch-Morell

**Affiliations:** 1Departamento Ciencias Biomédicas, Universidad CEU Cardenal Herrera, CEU Universities, 46115 Valencia, Spain; quiquegarciagen@gmail.com (E.G.-G.); mariola.penades@uchceu.es (M.P.); salvador.merida@uchceu.es (S.M.); carmen.desco@uv.es (C.D.); doctoranavea@retinavalencia.com (A.N.); 2Thematic Cooperative Health Network for Research in Ophthalmology (Oftared), Carlos II Health Institute, 28029 Madrid, Spain; 3Fundación para el Fomento de la Investigación Sanitaria y Biomédica de la Comunidad Valenciana (FISABIO), 46020 València, Spain; rafael.amvl@gmail.com

**Keywords:** high myopia, aqueous humor samples, complement factor H, choroid thickness, maculopathy

## Abstract

High myopia (HM) is both a medical problem and refractive error of the eye owing to excessive eyeball length, which progressively makes eye tissue atrophic, and is one of the main causes for diminishing visual acuity in developed countries. Despite its high prevalence and many genetic and proteomic studies, no molecular pattern exists that explain the degenerative process underlying HM, which predisposes patients to other diseases like glaucoma, cataracts, retinal detachment and chorioretinal atrophy that affect the macular area. To determine the relation between complement Factors H (CFH) and D (CFD) and the maculopathy of patients with degenerative myopia, we studied aqueous humor samples that were collected by aspiration from 122 patients during cataract surgery. Eyes were classified according to eyeball axial length as high myopia (axial length > 26 mm), low myopia (axial length 23.5–25.9 mm) and control (axial length ˂ 23.4 mm). The degree of maculopathy was classified according to fundus oculi findings following IMI’s classification. Subfoveal choroid thickness was measured by optical coherence tomography. CFH and CFD measurements were taken by ELISA. CFH levels were significantly high in the high myopia group vs. the low myopia and control groups (*p* ˂ 0.05). Significantly high CFH values were found in those eyes with choroid atrophy and neovascularization (*p* ˂ 0.05). In parallel, the CFH concentration correlated inversely with choroid thickness (R = −0.624). CFD levels did not correlate with maculopathy. All the obtained data seem to suggest that CFH plays a key role in myopic pathology.

## 1. Introduction

High myopia (HM) is both a medical problem and refractive error of the eye owing to excessive eyeball length, which progressively makes eye tissue atrophic, and is one of the main causes of diminishing visual acuity in developed countries [1,2]. High myopia prevalence is expected to rise in forthcoming years and to affect 938 million people by 2050, which makes it a first-order disease of clinical interest [3,4,5,6,7]. Despite its high prevalence and many genetic and proteomic studies, no molecular pattern exists that explains the degenerative process underlying HM, which predisposes patients to other diseases like glaucoma, cataracts, retinal detachment and chorioretinal atrophy [8,9,10,11]. 

High myopia can lead to blindness when severe chorioretinal atrophy develops. Another common ocular disease that can cause severe vision loss, is the final form of dry age-related macular degeneration (ARMD) called geographic atrophy. This ARMD variant comes with progressive choroid atrophy, followed by retinal atrophy and loss of vision. Although the two diseases do not seem to have the same ethiopathogenesis, to a certain extent, this is a similar process to progressive chorioretinal atrophy, which is typical of HM [12,13,14]. Alterations to complement factor H (CFH) and complement factor D (CFD) have been related to both conditions, ARMD and geographic atrophy. Former studies have evaluated how the complement system is involved in ARMD (age-related macular degeneration) and demonstrated how the association of the genes of the alternative pathway of the complement system, especially those involved in CFH and CFD, is more susceptible to ARMD [15,16,17,18,19]. The complement system is a critical component of the innate immunity responsible for the opsonization and removal of bacterial and apoptotic cell remains, and also for recruiting immune cells at infection sites and in tissue damage [20,21]. Its activation is triggered by several mechanisms depending on the involved pathway (classic, alternative and lectins). They all converge to form C3 convertase, which results in the formation of products C3a, C3b and C5a, and the C5b-9 membrane attack complex [20,21]. The efficiency of its functioning depends on the fine control of its activation. When it is deregulated or overactivated, it can perform homeostatic and pathological functions by driving different inflammatory disorders, which reflect the complement’s multifaceted nature [18,22].

Animal models have been valuable in investigating the mechanisms involved in the development of myopia. Numerous studies have shown that degradation of the visual input using form deprivation or lens defocus can induce axial myopia during the early postnatal period [23,24]. During this process, the retina is thought to initiate signal transduction triggered by blurred vision. The findings described in animal models suggest that both complement activation and inflammation plays a crucial role in the development of myopia. Moreover, the results provide novel evidence for transcriptional activation of the complement system during both myopia and hyperopia induction, and confirm existing literature implicating cell signaling, mitochondrial, and structural processes in refractive error. Further comparisons demonstrated association and extensive similarities in genes linked to age-related macular degeneration, choroidal neovascularization and cataract [15,25,26,27,28].

Nonetheless, the role played by the complement has not been studied to date in patients with HM and chorioretinal atrophy, although such knowledge may be very important for acquiring in-depth information about this pathology. So we have studied complement H and D factors in high myopic eyes in order to identify changes related to the presence or not of atrophy. 

## 2. Materials and Methods

We conducted a cross-sectional research study by selecting 122 eyes of patients who were being operated for cataracts at the FISABIO Medical Ophthalmological Clinic in Valencia (Spain) to analyze complement Factors H and D in the aqueous humor of myopic and control eyes. The inclusion criteria were: patients with cataracts ready to be operated, refractive hypermetropia defects and astigmatism below +0.75D and 1.75D, respectively. The exclusion criteria were: concomitant eye disease that could interfere with the results (ongoing maculopathy other than myopic maculopathy, uncontrolled glaucoma with two topical drugs, any stage of uveitis, venous or retinal arterial occlusions, hypermetropia refraction over +2D, astigmatism over 3D). All the participants in this study were Caucasian. They presented cataract eyes with nuclear color and opalescence, cortical or posterior subcapsular cataract 2–3, according to the Lens Opacities Classification System III. Uncontrolled high blood pressure with treatment, diabetes, high cholesterol index with treatment were considered the exclusion criterion. 

Experimental groups were formed according to eyeball axial length. Any eyes whose axial length exceeded 26 mm were classified as HM; axial lengths between 23.5 and 25.9 mm were classified as low myopia (LM); axial lengths below 23.4 mm were employed as the control group. As all the patients had cataracts, refractive status was not used to classify our patients in order to avoid any bias caused by index myopia [29]. 

This study was approved by the FISABIO Ethics and Clinical Research Committee, and the whole study complied with the Declaration of Helsinki. All the patients gave informed consent and underwent complete ophthalmological examination, including: ETDRS best-corrected visual acuity (BCVA); exploring the anterior segment by a slit-lamp; binocular ophthalmoscopy and wide-field retinography using Optos^®^ Optomap^®^ P200Tx to study fundus oculi; detecting detect presence or not of staphyloma; classifying maculopathy according to International Myopia Institute (IMI’s classification) [30]. The axial length measurements were taken by interferometry (Zeiss IOLMaster 700^®^) and optical coherence tomography (Swept-source optic coherence tomography SSOCT TOPCON, Tokyo, Japan) to obtain the choroidal thickness measurement. This measurement was manually taken with a measuring instrument (caliper) facilitated by the software that came with the apparatus. Measurements were performed by two different observers (CDE and RAM) masked for the axial length of the patient. Sections where foveal depression was well observed were selected, a measurement was made per patient and observer (two measurements for eye in total) at the point corresponding to the center of the fovea. The concordance rate was 95% between them.

Aqueous humor samples were collected during cataract surgery. After sterilizing eyelids and eyelashes, and instilling 5% iodine povidone on the conjunctival sac, a sterile adhesive dressing was placed to separate lashes. The eye surface was washed with saline solution, and paracentesis was performed at the site where the surgical incision was to be made using a 30 G needle to aspire an aqueous humor sample of 120 μL inside a 1cc syringe. Then surgery commenced following the standard technique. The sample hence collected was placed inside an eppendorf tube to be frozen in liquid nitrogen to be kept at −80 °C until it was used. 

The complement Factors H and D measurements were taken following the optimized protocol of the Human Adipsin ELISA Kit (catalog number: CSB-E14369h) and the Cusabio Human Complement Factor H ELISA Kit (catalog number: CSB-E08931h) (Cusabio Technology, Houston, TX 77054, USA). Statistical analyses were performed with version 24.0 of the commercially available IBM SPSS software (IBM Corp. 2016 for Windows, Version 24.0, Armonk, NY: IBM Corp., USA) and version 7.04 of GraphPad Prism for Windows (GraphPad Software, La Jolla, CA, USA). Data normality was verified by the Kolmogorv-Smirnov test (*p* > 0.05). The ANOVA of the data found by Levene’s test was performed by Tukey’s test as a *post hoc* analysis, provided that the homogeneity of variances was indicated (*p* ˂ 0.05). For those variables that did not present homogeneity of variances, the Kruskal-Wallis test was used as a non-parametric analysis. Pearson’s or Spearman’s correlations were employed to examine the strength of the association between two variables. The values of axial length, choroidal thickness and age were tested in a stepwise multiple linear regression to predict the Factor H and determine their individual effect on it. Statistical significance was set at *p* ˂ 0.05. Values are expressed as mean ± SD.

## 3. Results

The 122 patients were distributed as follows: 43 were the controls, 47 were LM and 32 were HM. Their mean age was 75.6 ± 6.9, 73.4 ± 10.3 and 66.4 ± 12.5 years, respectively, with 57% women and 43% men. Choroid thicknesses, as measured by optical coherence tomography, notably reduced in the HM group (128.4 ± 101.3 μm) compared to the control (237.1 ± 64.0 μm) and LM (199.2 ± 78.0 μm) groups, for whom significance was 0.01 (Table 1). There are no significant differences between both sexes, so the patients are treated as a single group. 

To analyze the developed pathologies in our studied patients, IMI´s classification was followed with five degrees of macular degeneration (Table 2). In 93.75% of the control patients, eyes looked normal, with tessellation in 6.25% of cases. Macular degeneration predominated more in the HM group, along with diffuse chorioretinal atrophy, lacquer cracks and macular atrophy in 85.18% of cases (Table 2). 

The analysis of CFH and CFD reflected significantly high mean concentrations (*p* ˂ 0.05) for CFH in the HM group (27.19 ± 15.47 ng/mL) compared to the LM (22.91 ± 10.52 ng/mL) and control (22.76 ± 5.90 ng/mL) groups (Figure 1a). The mean CFH values rose as eyeball axial lengths increased (Figure 1c) at the same time as this axial elongation reduced choroid thickness (Figure 1d). The mean CFD concentration obtained in the HM group (50.73 ± 12.62 ng/mL) was slightly lower than in the LM (53.20 ± 20.75 ng/mL) and control (52.81 ± 19.87 ng/mL) groups, with no significant differences (Figure 1b; *p* > 0.05). 

In accordance with the degree of developed maculopathy in patients, the choroid thickness analysis indicated a significant reduction in choroid thickness in the patients who presented lacquer cracks and neovascularization compared to the eyes that looked normal (Figure 2a). Axial length significantly increased in the patients showing lacquer cracks, choroidal atrophy and neovascularization in relation to the eyes that looked normal (*p* ˂ 0.05; Figure 2b). The Spearman’s correlation analysis performed between choroid thickness and CFH levels indicated a level of significance of *p* ˂ 0.05, and CFH correlated inversely with choroid thickness (R= −0.624) (Figure 2c). Complement Factor H presented higher levels as myopic maculopathy developed, and both macular atrophy and neovascularization significantly increased compared to the eyes that looked normal (*p* ˂ 0.05) (Figure 2d). 

We further found that both axial length and choroidal thickness were the factors associated with Factor H (β = −0.374, *p* < 0.001 and β = 0.253, *p* = 0.016, respectively), in the multiple regression model (*p* < 0.05). In this model, the contribution of age variable was insignificant (β = −0.048, *p* = 0.627). 

## 4. Discussion

Interest in the complement system has recently reappeared after its involvement in AMD has been discovered [14,22,31,32]. Among the factors that regulate the alternative pathway, we find Factor H, synthesized in the eye by the optic nerve, the sclera, the retinal pigment epithelium, the retina, the ciliary epithelium and the crystalline lens [33]. It inactivates Factor C3b directly to prevent cascade amplification. 

High myopia and developing myopic maculopathy are associated with an altered expression of vascular endothelial growth factors (VEGF) and pigment epithelium-derived (growth) factor (PEDF) [34,35,36], and with a higher regulation of hypoxia-induced factors [37]. In pathologic myopia, high C3a levels have been described in the plasma of patients with choroid angiogenesis compared to normal controls, which suggests an interaction between innate immunity and choroid angiogenesis [35,38]. Recent studies also describe how C3a acts by stimulating VEGF synthesis at the same time as PEDF synthesis lowers, which promotes the appearance of neovascularization [35]. 

The aqueous humor analysis is a very useful tool for not only studying the molecular mechanisms involved in axial lengthening, which is necessary for myopia progression, but also for understanding the complement’s role in HM because this would lead to new therapeutic strategies being developed [39,40]. In our article, we correlated the Factor H levels in aqueous humor with central myopia chorioretinal atrophy. To do so, we classified maculopathy according to fundus characteristics and the central choroid thickness measurement. We excluded all those patients with metabolic diseases that could interfere with the results. Thus, the alterations in FH levels were presumably due to ophthalmological alterations. We demonstrated an association between significantly high CFH levels and HM (Figure 1a) with significance set at *p* ˂ 0.05. Likewise, CFH correlated inversely with choroid thickness (R = −0.624). In turn, the CFH levels rose as maculopathy progressed, and significantly high levels were observed in those eyes presenting diffuse patchy atrophy, macular atrophy and plus disease versus the eyes classified as normal (Figure 2d). These were the two most notable contributions of our study because, as far as we know, no significant increase in Factor H associated with myopic choroidal atrophy has yet been described. Although our study did not allow us to state that this finding was due to a cause-effect reason, consequence or causal finding, the significantly high CFH levels in the HM group (*p* ˂ 0.05) could suggest a key role played for CFH in myopic pathology, althought to confirm this fact, it would be necessary to study it more deeply. This finding is surprising because, according to our results, CFH appeared to increase as disease progressed; that is, pigment epithelium atrophy was worse, which is associated with choroidal atrophy. Regarding the degree of choroidal atrophy, it must be considered that the study population was not homogeneous in terms of the age of the groups, the high myopic group had a clearly lower average age. This is because the study group was selected from among the patients who were operated on for cataracts. This fact could bias our results. However, we think that in the worst case it would suppose a positive bias, since at an older age a greater degree of choroidal atrophy is expected as is already described [41]. However, the HM group, being younger, has a greater degree of choroidal atrophy and at the same time greater alteration in the CHM than it would be expected in relation to age.

Lower subfoveal choroid thickness values have been recently associated with the presence of the CFH risk genotype and being more predisposed to chorioretininal pathologies. However, this has been described for populations older than the patients in our HM group, for whom such thinning occurred early [41]. The pigment epithelium is one of the main sources of this factor in eyes, which makes this finding paradoxical. One possible explanation is that this increase in HM is due to its synthesis by other eye structures that produce CFH: sclera, retina, optic nerve and ciliary epithelium [19,20,33]. This increased CFH concentration is spontaneously produced to block the action of Factor C3b by preventing the complement cascade from increasing [19,42]. 

There is evidence to suggest that oxidative stress exists in myopia, especially in HM [43,44,45]. Complement Factor H has antioxidant effects and regulates caspase-dependent apoptosis in retinal pigment epithelial cells submitted to oxidative stress. CFH also blocks the proinflammatory effects of malondialdehyde, a majority lipid peroxidation product, and protects from oxidative stress in vivo in mice [46,47]. In a recent study, our group evidenced nitrosative stress in the aqueous humor of HM patients. Tyrosine nitration is the only type of post-translational modification to occur in the inflammation and nitrosative stress context [48]. In ARMD, the presence of immunoreactive nitrotyrosine in CFH has been demonstrated, which promotes the secretion of proinflammatory and angiogenic cytokine IL-8 from monocytes stimulated with lipid peroxidation products. These findings clearly suggest that nitrated CFH contributes to ARMD progression [49] because the reduced form of CFH plays a protector role under oxidative/nitrosative stress conditions, and its oxidated form plays a pathological role as an activator of the alternative pathway [50]. Our findings agree with recent studies that have examined the complement’s involvement in geographic ARMD atrophy similarly to choroidal atrophy that develops in HM, and suggest that similar molecular systems might come into play to trigger such common symptoms [15,51]. 

Another complement factor that has been well-studied in relation to ARMD is Factor D. The role it plays as an activator of the alternative pathway is well-known. ComplementFactor D is one of the factors that activates the alternative pathway by acting directly on convertase C3 [20,21]. Although its pathogeny in ARMD remains unclear, evidence for factor D (FD) in the ARMD pathogeny is as follows: in a model of FD genetic deficiency, mice were unprotected against oxidative stress-mediated photoreceptors degeneration, and increased complement systemic activation, including Factor D, was detected in the serum of ARMD patients versus controls [18]. Our study found no significant differences in the mean complement Factor D concentrations in the HM, LM and control emmetropia groups [18,20], which agrees with the expected results as CFD acts as an activator of the alternative pathway, which is the opposite role to that played by CFH that acts as an inhibitor [52]. 

## 5. Conclusions

All the data herein obtained, which need to be subsequently verified, could suggest a key role for CFH in myopic pathology. However, based on available data, it is true that we cannot specify if the increases herein found could be due to a cause or a causality of myopic pathology. Further studies involving more patients will be necessary to deepen our understanding of the mechanisms behind this process. In any case, discovering how complement factors are involved in HM can lead to new therapeutic targets to address the development of more efficient pharmacological treatments [13,15,37,43,53,54,55,56]. 

## Figures and Tables

**Figure 1 jcm-10-02600-f001:**
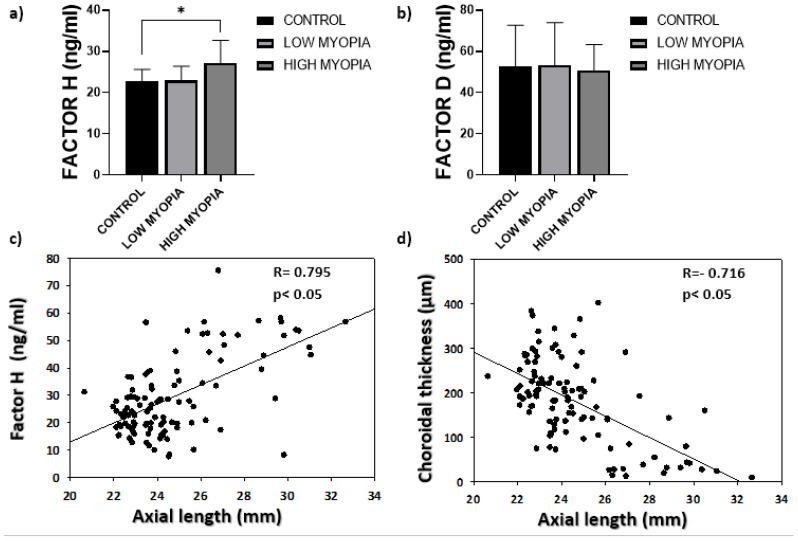
(**a**) The mean concentrations of complement Factor H. Values are expressed as mean ± SD. The difference in means (*) was established at a significance of *p* ˂ 0.05; (**b**) the mean complement Factor D concentrations; (**c**) the correlation analysis between eyeball axial length and the measured complement Factor H concentrations; (**d**) inverse correlation between eyeball axial length and choroid thickness measured by OCT.

**Figure 2 jcm-10-02600-f002:**
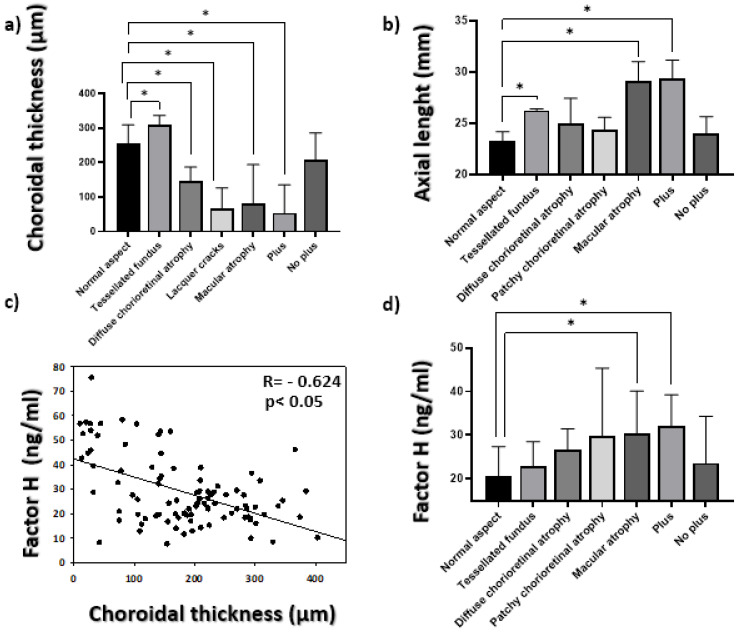
(**a**) The mean concentrations of choroidal thickness in each maculopathy. Values are expressed as mean ± SD; (**b**) Evolution of axial length and degree of maculopathy; (**c**) inverse between complement Factor H and choroid thickness measured by OCT (R= −0.624, *p* < 0.05); (**d**) mean values of correlation complement Factor H and degree of maculopathy. The difference in means (*) was established with a significance of *p* ˂ 0.05.

**Table 1 jcm-10-02600-t001:** Clinical characteristics of the 122 studied patients. Patients’ ethnicity was Caucasian. Groups were distributed as HM, LM and control for emmetropia according to axial length. Each value represents the mean ± SD, * *p* ˂ 0.05HM vs. Control group; # *p* ˂ 0.05 LM vs. Control group.

	N	AGE (years)	SPHERICAL EQUIVALENT	BCVA	AXIAL LENGTH (mm)	CHOROID THICKNESS (μm)
CONTROL	43	75.6 ± 6.9	−1.13 ± 1.55	0.35 ± 0.18	22.6 ± 0.4	237.1 ± 64.0
LM	47	73.4 ± 10.3	−3.4 ± 3.03	0.44 ± 0.37	24.1 ± 0.6 ^#^	199.2 ± 78.0
HM	32	66.4 ± 12.5	−9.69 ± 7.62	0.61 ± 0.53	28.1 ± 1.9 *	128.4 ± 101.3 *

**Table 2 jcm-10-02600-t002:** Distribution of maculopathies in the eyes of the studied patients following IMI’s classification expressed as % of the entire study population.

DEGREES OF MACULOPATHY	CONTROL	LM	HM
Normal aspect	20.49	17.21	0
TessellatED FUNDUS	0	0	1.63
dIFFUSE CHORIORETINAL ATROPHY	3.27	8.19	6.55
PATCHY ATROPHY	3.27	13.11	3.27
MACULAR ATROPHY	0	0.01	13.11
PLUS: NEOVASCULARIZATION/FUCHS PLUS: LACQUER CRAKS	0.01	0	12.29

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
