# Peer review of "High Myopia and the Complement System: Factor H in Myopic Maculopathy"

_jcm, 2021, doi:10.3390/jcm10122600_

Round 1
Reviewer 1 Report
This revised manuscript (original submission ID: jcm-1155186-v1) attempts to address the issues outlined by the reviewers. While the inclusion of a multivariate analysis helps to clarify some of the concerns of the original manuscript there are still a number of aspects of the paper that require further work.
The paper attempts to explain the similarities in complement factor H (CFH) levels being elevated in myopic maculopathy with those seen in geographic atrophy in age-related macular degeneration. The paper argues that the end stage of both disease processes is a subfoveal choroidal atrophy and the elevated CFH in both disease processes is evidence of its role in this atrophy. The authors have not adequately addressed the underlying disease processes to show a causative link between CFH and myopic maculopathy, and the results can only show a correlation. The disease process leading to geographic atrophy in ARMD is the result of drusen deposition in the sub-RPE space leading to degeneration of the RPE and subsequent choroidal atrophy. The pathogenesis of myopic maculopathy is tied to thinning and stretching of the tissue. To link these two processes together solely based on elevated CFH levels and an end-stage of choroidal atrophy appears tenuous. No evidence has been provided to show elevated CFH is causative of the changes in myopic maculopathy and not a by-product of the degenerative process. While the inability to determine cause and effect is mentioned in the discussion, the link with the changes observed in ARMD is given too much weight in the introduction and discussion and is not justified by the results.
The multiple regression analysis shows the results are not correlated with the age of participants, which is good to see. The assertion, however, that axial length and choroidal thickness are independent factors associated with CFH needs to be justified. How can you claim that both factors are independently associated with CFH when, as clearly observed in Figure 1d, there is a significant negative correlation between axial length and choroidal thickness? Thinning of the choroid is not independent of axial length, and indeed thinning of the choroid in high myopia is correlated with increasing axial length. This interdependence of choroidal thickness and axial length means that the two factors cannot both be independently correlated with CFH. Further discussion of the covariate nature of these findings needs to occur.
In the response to the reviewers it is stated that as the analysis was “focused on severe macular atrophy as an expression of the extreme decrease of choroidal thickness, the classic and simple Avila classification seemed adequate to us” compared to more recent myopic maculopathy classification systems. This is in contrast to figure 2 and the discussion, where you focus upon the levels of CFH in different classifications of pathological changes with high myopia. You present CFH levels segregated by tessellation, posterior staphyloma, lacquer cracks, choroidal atrophy, and neovascularisation. In the discussion you argue that elevated CFH levels in eyes with choroidal atrophy and neovascularisation show that “CFH levels rose as maculopathy progressed”. In the META-PM classification, adopted by the International Myopia Institute, lacquer cracks and choroidal neovascularisation are classified as “Plus” lesions as they can occur at any stage of myopic maculopathy. The argument, therefore, that elevated CFH levels shows a correlation with increased progression does not hold for eyes with neovascularisation (or indeed that lacquer cracks are an indication of lower levels of myopic maculopathy) as it can occur with any level of myopic maculopathy. A more nuanced analysis of increasing maculopathy level is therefore required than that offered in the current analysis.
Fairly strong conclusions about increasing CFH levels and myopic maculopathy are made in the discussion. However, when the number of participants who displayed choroidal atrophy is examined (in Table 2), it can be seen that only 7 individuals have this level of maculopathy. Despite this, the pooled CFH levels for all high myopes, regardless of maculopathy status, is used when comparing high myopia to ‘normal’ and low myopia groups. This potentially overestimates the role of CFH in pathology development, given lacquer cracks (6 individuals) and neovascularisation (5 individuals), pathologies that can occur at any level of myopic maculopathy, have a higher proportional contribution to the CFH levels in the high myopia group. This again highlights the potential disadvantage of using the Avila classification over the META-PM classification of myopic maculopathy, given the Avila classification provides a higher weighting to the severity of the maculopathy to lacquer cracks and neovascularisation than the staged chorioretinal atrophy approach used in the META-PM classification. A discussion of the advantages and disadvantages of the classification system used must therefore have greater representation in the methods and discussion.
Reviewer 2 Report
Dear Authors,
You have corrected well.
Thanks,
Author Response
There is no suggestions
Round 2
Reviewer 1 Report
This revised manuscript positions the research much better in terms of the potential role for complement in the pathology of myopic macular degeneration. I believe the changes made to the manuscript strengthen the arguments, and no longer over-state the findings.
The revised introduction and its description of the possible link between myopia and complement, and how this potentially ties in with the changes in ARMD, is much clearer. I think this greatly improves the hypothesis being examined and more clearly explores the underlying mechanisms than previously. The discussion also better addresses the outcomes. Thank you.
The use of the IMI classification for myopic degeneration makes the argument more convincing of a link between complement factor H and the observed degenerations. Thank you for reanalysing the data with the new classification. I believe the paper is stronger for this updated analysis.
A few minor suggested corrections:
Introduction, Page 1, Line 36: Sentences should not start with an abbreviation. Change to “High myopia”.
Introduction, Page 1, Line 45: “There are another common…” should be shortened to “Another common…”
Methods, Page 3, Line 114: Please provide the full definition of IMI, International Myopia Institute, at the first use of this abbreviation.
Methods, Page 3, Line119: There is a second full stop after “apparatus” that should be removed.
Methods, Page 3, Lines 144-145: The font used in the sentence “The values of axial length…” is different from the rest of the text.
Results, Page 4, Table 2: Value for HM Macular Atrophy not aligned correctly. Also, in the table legend, please expand on what percentage refers to here (i.e. “expressed as percentage (%) of the entire study population”).
Results, Page 5, Line 189: Sentences should not start with an abbreviation. Change to “Complement factor H”.
Discussion, Page 6, Line 208: Missing “lens” after “crystalline”.
Discussion, Page 6, Lines 209-216: Given the negative results for Factor D, I would suggest moving these lines to the end of the discussion (Page 8, Lines 287-293). It seems to me that this would be a better fit for what is being discussed.
Discussion, Page 6, Line 212: Factor D has already been abbreviated in line 210 above, so no need to duplicate this here.
Discussion, Page 7, Line 217: Sentences should not start with an abbreviation. Change to “High myopia”.
Discussion, Page 8, Line 270-271: There is highlighting of the text that should be removed.
Discussion, Page 8, Line 271: Sentences should not start with an abbreviation. Change to “Complement factor H”.
References: Please check all references to ensure full bibliographical detail and appropriate journal formatting are provided for all references. I noted the following errors, but there may be others that need correcting:
References, Page 9, Line 345: Journal title needs italics.
References, Page 9, Lines 350-351: This reference is a duplication.
References, Page 10, Lines 379-380: Full bibliographic details of the journal need to be included.
References, Page 10, Lines 389-390: Which journal?
References, Page 11, Lines 417-418: Which journal?
References, Page 11, Lines 424-425: Issue and page numbers?
References, Page 11, lines 426-427: Issue and page numbers?
References, Page 11, Line 430: Full bibliographic information for a Book chapter should be provided.
References, Page 12, Lines 452-453: Issue and page numbers?
Author Response
Introduction, Page 1, Line 36: Sentences should not start with an abbreviation. Change to “High myopia”. Done
Introduction, Page 1, Line 45: “There are another common…” should be shortened to “Another common…” Done
Methods, Page 3, Line 114: Please provide the full definition of IMI, International Myopia Institute, at the first use of this abbreviation. Done
Methods, Page 3, Line119: There is a second full stop after “apparatus” that should be removed. Done
Methods, Page 3, Lines 144-145: The font used in the sentence “The values of axial length…” is different from the rest of the text.Done
Results, Page 4, Table 2: Value for HM Macular Atrophy not aligned correctly. Also, in the table legend, please expand on what percentage refers to here (i.e. “expressed as percentage (%) of the entire study population”). Done
Results, Page 5, Line 189: Sentences should not start with an abbreviation. Change to “Complement factor H”. Done
Discussion, Page 6, Line 208: Missing “lens” after “crystalline”. Done
Discussion, Page 6, Lines 209-216: Given the negative results for Factor D, I would suggest moving these lines to the end of the discussion (Page 8, Lines 287-293). It seems to me that this would be a better fit for what is being discussed. Done
Discussion, Page 6, Line 212: Factor D has already been abbreviated in line 210 above, so no need to duplicate this here. Done
Discussion, Page 7, Line 217: Sentences should not start with an abbreviation. Change to “High myopia”. Done
Discussion, Page 8, Line 270-271: There is highlighting of the text that should be removed. Done
Discussion, Page 8, Line 271: Sentences should not start with an abbreviation. Change to “Complement factor H”. Done
References: Please check all references to ensure full bibliographical detail and appropriate journal formatting are provided for all references. I noted the following errors, but there may be others that need correcting: Done
References, Page 9, Line 345: Journal title needs italics. Done
References, Page 9, Lines 350-351: This reference is a duplication. Done
References, Page 10, Lines 379-380: Full bibliographic details of the journal need to be included. Done
References, Page 10, Lines 389-390: Which journal? Done
References, Page 11, Lines 417-418: Which journal? Done
References, Page 11, Lines 424-425: Issue and page numbers? Done
References, Page 11, lines 426-427: Issue and page numbers? Done
References, Page 11, Line 430: Full bibliographic information for a Book chapter should be provided. Done
References, Page 12, Lines 452-453: Issue and page numbers? Done

This manuscript is a resubmission of an earlier submission. The following is a list of the peer review reports and author responses from that submission.
Round 1
Reviewer 1 Report
I think this report is worth at the point that this would be the first study that suggests there is a kind of relationship between CFH and myopic maculopathy. However, I have some opinions about the contents.
First, why did you separate participants, as normal, low myopia, and high myopia? If you wanted to show linear relationship between CFH levels and myopic grade, you have to analyze it.
Second, you explained about the association between CFH and myopic maculopathy using the relationship of oxidative stress and AMD, it could not be correct because the diseases have different pathologies.
Third, if you wanted to demonstrate there was a significant relationship between CFH and myopic maculopathy, you have to analyze it adjusted by axial length or choroidal thickness, because you showed that associations of CFH and AL or choroidal thickness. Additionally, you had better perform a linear trend test between CFH levels and myopic maculopathy grade.
Lastly, I found some minor mistakes in the text and figures.
p1 Line45 can led →can lead
Figure 2.a) Figure legend and the figure do not match.
p7 Line245 double comma
Line 258-264 There is the same texts as conclusions.
p9 Reference Line 373-374 no title of the journal
Reviewer 2 Report
This paper investigated the potential role of the complement system and inflammation on the development of myopic maculopathy in an adult human population. The study found a correlation between complement factor H levels and axial length, and an inverse correlation with choroidal thickness. There are a number of aspects in the paper that would benefit from further clarification.
No analysis of the age of the participants in the three groups has been made. The mean age of the control group is 9 years older than in the high myopia group, but no statistical analysis showing whether or not this was a significant difference has been presented. Therefore, a change in complement factor levels with age rather than refractive status cannot be ruled out. Furthermore, in the results a significant correlation between axial length and choroidal thickness is shown, along with significant correlations between Factor H and both axial length and choroidal thickness. The individual analyses that have been performed between the different components is not conducive to understanding which aspect is driving these differences. While the inability to determine a causative factor is acknowledge in the discussion, a more inclusive statistical analysis might enable more detailed conclusions to be drawn. I would suggest that a multivariate analysis, including age, axial length, choroidal thickness, and complement factor H would be appropriate to see how these factors interrelate.
Detail of how the choroidal thickness was measured needs to be added to the methods. Did the same observer make all the choroidal thickness measurements? How was centration in the middle of the fovea confirmed (i.e. the measures were made at the “point corresponding to foveal depression”, but how was alignment of the b-scan to the centre of the fovea confirmed)? Was the observer masked to the refractive status of the patient? Given the manual segmentation process of determining the choroidal thickness, what was the repeatability of the measurements? Was a single measure of choroidal thickness made for each patient or do the choroidal thickness measures represent an average of multiple measures?
The cataract in the sample population is described as “nuclear color and opalescence, cortical or posterior subcapsular cataract 2-3”. Why was no analysis of the type and severity of cataract conducted? Did cataract type/severity influence the complement levels? Given the link between myopia and cataract formation, and the younger age of the high myopia cohort compared to the control group, could the differences in complement level lead to the cataract formation or occur secondarily to the cataract formation?
The choice of definitions for staphyloma and myopic maculopathy have not been justified. The staphyloma definition chosen was from Curtin from 1977. While this represents the classical staphyloma definition, more recent studies have shown that the structural correlates do not necessarily fit with the fundoscopy image classification detailed in Curtin’s original paper. A more recent classification, correlating MRI-based structural changes to fundus images (2014; https://pubmed.ncbi.nlm.nih.gov/24813630/) would be more appropriate. Likewise, the classification of myopic maculopathy by Avila is from 1984, while there are more recent classification systems like the META-PM (2015; https://pubmed.ncbi.nlm.nih.gov/25634530/) or ATN (2019; https://pubmed.ncbi.nlm.nih.gov/30391362/) definitions. What is the justification for using these older classification systems, which are not in keeping with the proposed classifications for clinical studies from the International Myopia Institute (https://iovs.arvojournals.org/article.aspx?articleid=2727312)?
Title: “The high myopia” should just be “High myopia” in the title.
Introduction: It would be good to include discussion of animal models of myopia that also investigate complement/inflammatory mediators in the introduction and discussion (e.g. https://www.ncbi.nlm.nih.gov/pmc/articles/PMC4539642/ , https://pubmed.ncbi.nlm.nih.gov/27470424/)
Methods, Page 3, Line 101: If the definition of staphyloma that is used continues to be that of Curtin, then the reference needs to include the date here, and the complete reference in the reference list (Transactions of the American Ophthalmological Society, 1977; 75: 67-86).
Methods, Page 3, Line 101-102: If the definition of myopic maculopathy that is used continues to be that of Avila, et al., the reference needs to be included here, rather than in the Results, line 142.
Discussion, Page 6, Line 179: Outline what these “certain diseases” are.
Conclusion, Page 7: This is just a duplication of the last paragraph of the discussion. Delete and move conclusion heading up.